# Mapping Forest Growing Stem Volume Using Novel Feature Evaluation Criteria Based on Spectral Saturation in Planted Chinese Fir Forest

Hui Lin [1,2,3], Wanguo Zhao [1,2,3], Jiangping Long [1,2,3,*], Zhaohua Liu [1,2,3], Peisong Yang [1,2,3], Tingchen Zhang [1,2,3], Zilin Ye [1,2,3], Qingyang Wang [4] and Hamid Reza Matinfar [5]

[1] Research Center of Forestry Remote Sensing and Information Engineering, Central South University of Forestry and Technology, Changsha 410004, China
[2] Key Laboratory of Forestry Remote Sensing Based Big Data and Ecological Security for Hunan Province, Changsha 410004, China
[3] Key Laboratory of State Forestry Administration on Forest Resources Management and Monitoring in Southern Area, Changsha 410004, China
[4] Department of Land Surveying and Geo-Informatics, The Hong Kong Polytechnic University, Hung Hom, Kowloon, Hong Kong, China
[5] Soil Science Department, Lorestan University, Khoramabad 68151-44316, Iran
[*] Correspondence: longjiangping@csuft.edu.cn; Tel.: +86-0731-8562-3848

**Abstract:** Forest growing stem volume (GSV) is regarded as one of the most important parameters for the quality evaluation and dynamic monitoring of forest resources. The accuracy of mapping forest GSV is highly related to the employed models and involved remote sensing features, and the criteria of feature evaluation severely affect the performance of the employed models. However, due to the linear or nonlinear relationships between remote sensing features and GSV, widely used evaluation criteria inadequately express the complex sensitivity between forest GSV and spectral features, especially the saturation levels of features in a planted forest. In this study, novel feature evaluation criteria were constructed based on the Pearson correlations and optical saturation levels of the alternative remote sensing features extracted from two common optical remote sensing image sets (GF-1 and Sentinel-2). Initially, the spectral saturation level of each feature was quantified using the kriging spherical model and the quadratic model. Then, optimal feature sets were obtained with the proposed criteria and the linear stepwise regression model. Finally, four widely used machine learning models—support vector machine (SVM), multiple linear stepwise regression (MLR), random forest (RF) and K-neighborhood (KNN)—were employed to map forest GSV in a planted Chinese fir forest. The results showed that the proposed feature evaluation criteria could effectively improve the accuracy of estimating forest GSV and that the systematic distribution of errors between the predicted and ground measurements in the range of forest GSV was less than 300 $m^3/hm^2$. After using the proposed feature evaluation criteria, the highest accuracy of mapping GSV was obtained with the RF model for GF-1 images ($R^2$ = 0.49, rRMSE = 28.67%) and the SVM model for Sentinel-2 images ($R^2$ = 0.52, rRMSE = 26.65%), and the decreased rRMSE values ranged from 1.1 to 6.2 for GF-1 images (28.67% to 33.08%) and from 2.3 to 6.8 for Sentinel-2 images (26.85% to 33.28%). It was concluded that the sensitivity of the optimal feature set and the accuracy of the estimated GSV could be improved using the proposed evaluation criteria (less than 300 $m^3/hm^2$). However, these criteria were barely able to improve mapping accuracy for a forest with a high GSV (larger than 300 $m^3/hm^2$).

**Keywords:** forest growing stem volume; spectral saturation; feature evaluation criterion; kriging spherical model; quadratic model

## 1. Introduction

Forest growing stem volume (GSV), widely applied in forest resource investigations, is regarded as one of the most important indices in forest resource quality evaluation [1–3].

Traditional approaches of mapping forest GSV need a lot of field investigation [4–6]. In recent years, remote sensing technology has become one of the most important methods used to estimate forest parameters, such as forest height, GSV and above ground biomass (AGB) [7,8]. In particular, optical remote sensing images with rich data resources are widely applied to evaluate forest types and quantitatively estimate forest parameters [9,10]. Normally, optical remote sensing images reflect the forest growth process and changes in the forest canopy information, and the spectral variables extracted from optical image sets, including Landsat, Sentinel-2, GF-1, and GF-2, are often used to construct models for mapping GSV. However, these variables are insensitive to changes in GSV when the GSV is larger than the saturation levels. Previous studies also proved that the accuracy of mapping GSV using optical images is affected by an underestimated GSV because of the saturation phenomenon [11,12]. Therefore, it is still difficult to obtain reliable GSV using optical images in plantation forests with high GSV levels.

To improve the accuracy of GSV mapping, underestimated GSV should be corrected by delaying saturation levels by selecting appropriate feature sets [13]. Generally, it has been proved that a forest's vertical structure parameters are more sensitive to GSV than forest canopy information [14]. Previous studies have explored various methods to extract vertical forest parameters by delaying the saturation phenomenon, such as synthetic aperture radar (SAR) or polarimetric SAR data, airborne or spaceborne light detection and ranging (LiDAR), and CHM products generated from stereo image pairs [15–18]. Though some results have showed that saturation levels can be delayed by adding vertical forest parameters, airborne LiDAR is not suitable for assessing forest height over large regions because of the high cost of acquiring data [19,20]. In addition, for orbital LiDAR data, the accuracy of forest height in light spots rarely meets the requirements for forest parameter estimation in medium-scale regions. Therefore, optical remote sensing images remain the main source of data for mapping forest parameters.

Furthermore, feature evaluation criteria are also important for improving underestimated or overestimated results in mapping forest GSV [21–23]. Normally, optical variable sets are rather dependent on the employed evaluation metrics, and the accuracy of estimated GSV is also highly related to the sensitivity of selected variables [24]. Generally, feature evaluation methods are mainly based on the linear or nonlinear relationships between remote sensing features and GSV, such as importance, distance correlation coefficient (DC), maximal information coefficient (MIC), and Pearson correlation coefficient [7,25–27]. However, these inadequately express the sensitivity between forest GSV and spectral features, especially for the planted forest with a high GSV. After all, underestimated forest GSVs are mainly caused by the spectral saturation phenomenon that occurs when using optical remote sensing images.

Additionally, optimal feature sets are obtained with various feature selection methods. Commonly, the feature selection methods can be broadly classified into three categories: filters, wrappers, and embedded. Filters are entirely based on a single evaluation criterion, such as importance or Pearson correlation. However, determining this criterion and its threshold is rather difficult for various sensors and features [28,29]. Wrappers combine the feature selection methods with training models, and then, an optimal feature set is extracted with a certain accuracy index and repeatedly trained models [30,31]. However, these feature selection methods only refer to a single criterion, such as linear independence, information entropy, or precision index, and the saturation level of each variable is not considered as an evaluating feature method [32,33]. Therefore, to correct an underestimated GSV, the saturation levels of alternative variables should be viewed as a criterion to evaluate the sensitivity between the variables and GSV [34–36]. In addition, because the kriging spherical model and the quadratic model are gradually being applied to quantitatively evaluate the saturation levels of the features extracted from remote sensing images, it is more advantageous to focus on spectral saturation in feature evaluation and selection [37,38].

To overcome the disadvantages of traditional feature evaluation criteria in mapping forest GSV, we constructed novel feature evaluation criteria based on the Pearson correlations and optical saturation levels of each alternative remote sensing feature. In this study, two widely used optical remote sensing image sets, GF-1 and Sentinel-2, were acquired from a planted Chinese Fir Forest, and several types of common features, bands, vegetation indices, and texture features with different sizes were extracted from the acquired images. Then, the saturation level of each extracted feature was estimated using the kriging spherical model and the quadratic model. Next, the proposed feature evaluation criteria were constructed and applied to obtain optimal feature sets for mapping forest GSV with four models: the support vector machine (SVM), multiple linear regression (MLR), random forest (RF), and K-nearest neighbor (KNN) models. The capability of the new feature evaluation criteria was further evaluated with accuracy indices for mapping forest GSV.

## 2. Study Area and Data

### 2.1. Study Area

In this study, the Huangfengqiao state-owned forest farm (113°04′ to 113°43′E, 27°06′ to 27°24′N), located in You County, Hunan province, China, was selected as the study area. The landform of the forest farm is mainly low and middle mountains, with the elevation ranging from 115 m to 1270 m and the slope ranging from 20° to 35°. The climate type of the region is a subtropical monsoon humid climate, and the annual mean temperature and the average annual rainfall are 17.8 °C and 1410.8 mm, respectively. The forest coverage rate reaches 86.24% in the region (10,122.6 ha) and comprises Chinese fir (*Cunninghamia Lanceolata*), Pinus massoniana Lamb, bamboo, and Liriodendron chinense. The planted Chinese fir is the dominant species in the study area (Figure 1).

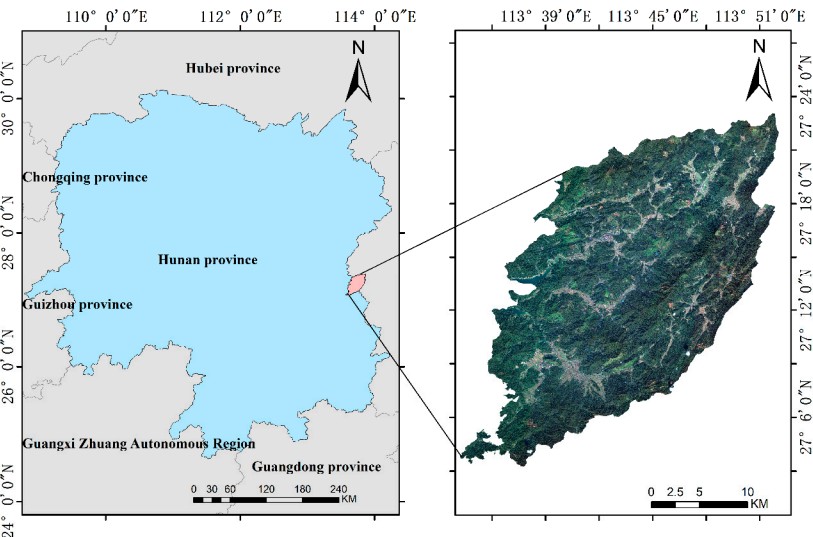

**Figure 1.** Location map of the study area.

### 2.2. Data Preparation

(1)    Ground Data

Because of the irregular distribution of the planted Chinese fir forest, a random stratification sampling method was used to obtain the distribution of the ground samples. Then, a total of 52 Chinese fir plots were measured during field investigations in 2016 and 2017 (Figure 2a). For all ground samples, the size of each sample (Including 19 plots of 20 m × 20 m and 33 plots of 30 m × 30 m) was determined by the complexity of the terrain. Moreover, the positions of the corner and central points of samples were accurately measured using a global positioning system (GPS) (positioning error of less than 10 cm). In each sample, parameters related to forest GSV were measured in ground field work, tree height was measured using a laser altimeter, diameter at breast height (DBH) was measured

with a diameter ruler, and crown width in the east–west and north–south directions was measured with a tape. The stem volume of each tree was calculated based on the binary volume table of planted Chinese fir in Hunan province (Equation (1)), and the GSV of each sample was the sum of all tree stem volumes within the plot. In the study, the maximum DBH and height were 29.48 cm and 20.5 m, respectively. The GSV of the measured samples ranged from 59 $m^3/hm^2$ to 480 $m^3/hm^2$, and relationships between forest GSV and DBH are plotted in Figure 2b.

$$V = a \times DBH^b \times H^c \tag{1}$$

where $V$ is the volume of each tree, $a = 0.000058777042$, $b = 1.9699831$, and $c = 0.89646157$.

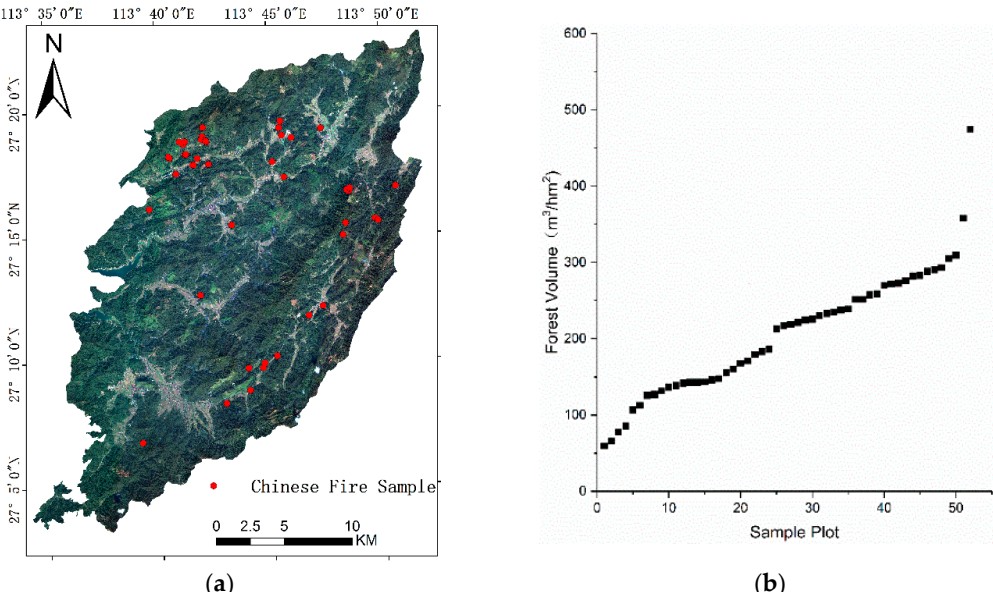

|  (a)  |  (b)  |

**Figure 2.** Spatial distribution of samples. (**a**) The map of ground samples; (**b**) the range of measured GSV.

(2)　　Remote sensing data and pre-processing

　　In this study, two remote sensing image sets, GF-1 and Sentinel-2, were employed to map the forest GSV in a planted Chinese fir forest. Specially, GF-1 images with 4 multispectral bands (spatial resolution: 8 m) and 1 panchromatic band (spatial resolution: 2 m), were acquired on 29 July 2016 (Table 1). Meanwhile, Sentinel-2 images with 13 narrow bands were acquired on 1 November 2017, and the spatial resolution of these bands was 10 m, 20 m and 60 m [39]. To extract alternative variables from these remote sensing images, a series of pre-processing steps, including radiometric calibration, atmospheric correction, ortho-correction, geometric correction, image fusion, and clipping, were applied to the GF-1 and Sentinel-2 images [40]. The preprocessing of remote sensing data was implemented in ENVI 5.3 software. Ultimately, these pre-processed images were resampled to 30 m to reduce the matching error between images and samples.

**Table 1.** The information of acquired GF-1 and Sentinel-2 images.

| Remote Sensing Data | Acquisition Time | Band | Wavelength/(μm) | Resolution/(m) |
|---|---|---|---|---|
|  |  | Blue | 0.45–0.52 | 8 |
|  |  | Green | 0.52–0.59 | 8 |
| GF-1 | 29 July 2016 | Red | 0.63–0.69 | 8 |
|  |  | Near infrared | 0.77–0.89 | 8 |
|  |  | Panchromatic | 0.45–0.89 | 2 |

**Table 1.** *Cont.*

| Remote Sensing Data | Acquisition Time | Band | Wavelength/(μm) | Resolution/(m) |
|---|---|---|---|---|
| Sentinel-2 | 1 November 2017 | Blue | 0.490 | 10 |
| | | Green | 0.560 | 10 |
| | | Red | 0.665 | 10 |
| | | Vegetation Red Edge | 0.705 | 20 |
| | | Vegetation Red Edge | 0.740 | 20 |
| | | Vegetation Red Edge | 0.783 | 20 |
| | | NIR | 0.842 | 10 |
| | | Vegetation Red Edge | 0.865 | 20 |
| | | SWIR | 1.610 | 20 |
| | | SWIR | 2.190 | 20 |

## 3. Methods

### 3.1. Extracting Features

After image preprocessing, three types of features (spectral bands, vegetation indices, and texture features) were extracted from GF-1 and Sentinel-2 images (Table 2). In our study, the number of spectral bands was four for GF-1 images and eleven for Sentinel-2 images. Additionally, several texture features with different sizes were extracted from each spectral band using a gray-level co-occurrence matrix (GLCM). Considering the difference in spatial resolutions, the sizes of the extracted texture features ranged from $3 \times 3$ to $19 \times 19$. Moreover, several vegetation indices were also extracted from these images [41,42].

**Table 2.** Three types feature variables extracted from GF-1 and Sentinel-2 images.

| Variable Type | Description | Reference |
|---|---|---|
| Spectral bands | GF-1: Blue, Green, Red, NIR | [33] |
| | Sentinel-2: Blue, Green, Red, VRE1, VRE2, VRE3, NIR, Narrow NIR, Water Vapor, SWIR1, SWIR2 | [37] |
| Vegetation indices | $SAVI = (1 + L) \times (NIR - RED)/(NIR + RED + L)(L = 0.5)$ | [43] |
| | $ARVI = [NIR - (2 \times RED - BLUE)]/[NIR + (2 \times RED - BLUE)]$ | [44] |
| | $EVI = 2.5 \times (NIR - RED)/(NIR + 6 \times RED - 7.5 \times BLUE + 1)$ | [45] |
| | $TVI = \sqrt{\frac{NIR-RED}{NIR+RED} + 0.5}$ | [31] |
| | $MSR = (NIR/RED - 1)/(\sqrt{\frac{NIR}{RED}}+1)$ | [22] |
| | $NLI = (NIR2 - RED)/(NIR2 + RED)$ | [30] |
| | $DVIij = Bandi - Bandj$ | [37] |
| | $RVIij = Bandi/Bandj$ | [23] |
| | $NDVIij = (Bandi - Bandj)/(Bandi + Bandj)$ | [14] |
| | $NDVIijk = (Bandi + Bandj - Bandk)/(Bandi + Bandj + Bandk)$ | |
| Texture features | Mean, Variance (VAR), Homogeneity (HOM), Contrast (CON), Dissimilarity (DIS), Entropy (ENT), Second Moment (SM), Correlation (COR) | [46] |

### 3.2. Spectral Saturation and Estimation Model

In previous studies, saturation levels were indirectly obtained from extreme values based on scatterplots or fitted curves between selected variables and GSV via visual interpretation. Over the past ten years, some models related to saturation levels have been applied to determine saturation levels [46–48]. In particular, a semi-exponential model has often been employed to estimate saturation levels using SAR or polarimetric SAR images, and the saturation level can be directly extracted from parameter models after solving them [18,43–45]. For optical images, a kriging model based on the semi-covariance function in geo-statistics has been widely applied to quantitative analysis spectral saturation. Moreover, it has also been proven that the quadratic model can characterize spectral saturation

levels. Therefore, the kriging and quadratic models were applied to evaluate the saturation levels in this study.

(1) Kriging Model

When the GSV reaches a certain value, the saturation of spectral variables will occur, and the saturation phenomenon has been proved to be similar to the distribution of spatial autocorrelation in geo-statistics [14,38,45]. The kriging model with a spherical function has the capability of describing the phenomenon of spectral saturation related to GSV. In this study, the kriging model with a spherical function was applied to fit the relationships between GSV and the selected variables:

$$y(h) = \begin{cases} c_0 & h = 0 \\ c_0 + c\left(\frac{3h}{2a} - \frac{h^3}{2a^3}\right) & 0 < h \le a \\ c_0 + c & h > a \end{cases} \tag{2}$$

where $y$ is the selected spectral variable, $h$ is the value of the GSV, $c_0$ is the nugget parameter indicating the spectral reflectance value at $h = 0$, $c$ is arch height indicating the change rates of the spectral reflectance of the selected variable and the GSV, $c_0 + c$ is the maximum or minimum spectral reflectance when the GSV reaches its saturation value, $a$ is the value of saturation related to GSV; when $-\frac{c^3}{2a^3} = a_2$, $\frac{3c}{2a} = a_1$, $c_0 = a_0$, $x_1 = h$, and $x_2 = h_3$, the spherical model is linearized and its coefficients of linear regression can be obtained using least squares regression:

$$y = a_2 \times x_2 + a_1 \times x_1 + a_0 \tag{3}$$

After resolving the linearized model, the saturation level located at the extreme value of the function is calculated according to the model parameters. Combined with the extreme point of the curve of the model, the saturation of the kriging spherical model is expressed as:

$$saturation\ values = \sqrt{-\frac{a_1}{3a_0}} \tag{4}$$

(2) Quadratic model

The quadratic model has been applied to fit the curves between the GSV and spectral variables. The quadratic model is expressed as follows:

$$y = a_0 x^2 + a_1 x + a_2 \tag{5}$$

where $y$ is the selected spectral variable, $x$ is the value of the GSV, and $a_0$, $a_1$, and $a_2$ are model parameters. After solving the model, the extreme point of the fitted curve is regarded as the saturation value. Therefore, the saturation of the quadratic model is expressed as:

$$saturation\ values = -\frac{a_1}{2a_0} \tag{6}$$

Based on the spherical and quadratic models, the saturation values and the curves between the estimated GSV and selected spectral variable are illustrated in Figure 3 (band 3 for GF-1 in Figure 3a and band 5 for Sentinel-2 in Figure 3b). It is shown that the estimated saturations of band 3 extracted from GF-1 was 289.68 $m^3/hm^2$ for the spherical model and 332.1 $m^3/hm^2$ for the quadratic model. Additionally, the estimated saturation of band 5 extracted from Sentinel-2 was 266.25 $m^3/hm^2$ for the spherical model and 299.24 $m^3/hm^2$ for the quadratic model. Due to the differences in the spectral saturation estimation models, the estimated saturation of the same spectral feature was different.

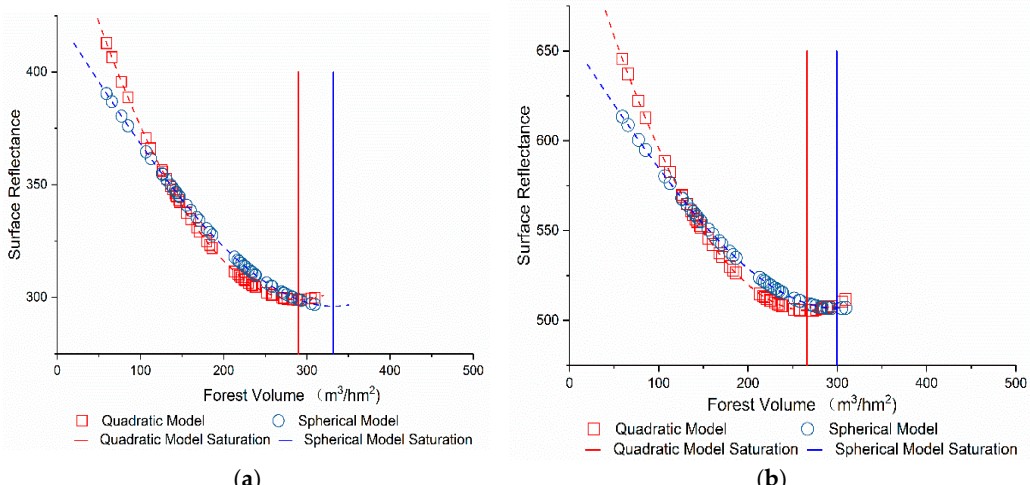

**Figure 3.** GSV saturation diagram based on different remote sensing features. (**a**) Band 3 from GF-1; (**b**) band 5 from Sentinel-2.

### 3.3. Feature Selection Method Based on Spectral Saturation

Normally, the accuracy of mapped forest GSV is directly dependent on the sensitivity of the selected feature set. However, the Pearson correlation coefficient is widely regarded as an important index used to evaluate sensitivity based on linear independence. Moreover, the spectral saturation levels of remote sensing features can directly describe the range of estimated forest GSV. Therefore, the saturation levels of features should also be regarded as an index to evaluate sensitivity. Theoretically, the selected features in an optimal variable set should have high spectral saturation levels and Pearson correlations. Therefore, new feature evaluation criteria were proposed based on the spectral saturation and Pearson correlation coefficient. The steps of the proposed feature evaluation criteria are as follows:

(1) The Pearson correlation coefficient and spectral saturation levels of all extracted remote sensing features related to GSV are first obtained and sorted in descending order.

(2) Based on the sorted Pearson correlation coefficient and spectral saturation levels, the serial numbers $r_{1i}$ (Pearson correlation coefficient) and $r_{2i}$ (spectral saturation levels) of each feature $i$ are obtained.

(3) To obtain the initial variable set, a multiple linear stepwise regression model and an adjusted determination coefficient are employed to determine the number of features sorted by the Pearson correlation coefficient and spectral saturation levels, respectively. The initial variable set is determined based on the maximum values of the adjusted determination coefficients $R_1$ (Pearson correlation coefficient) and $R_2$ (spectral saturation levels).

(4) In an initial variable set, a new importance evaluation index (PS: Pearson and saturation) is constructed by using the serial numbers and the determination coefficient with the following formula:

$$PS_i = \frac{r_{1i}}{R_1} + \frac{r_{2i}}{R_2} \tag{7}$$

where $PS_i$ is the value of the importance evaluation index corresponding to the Pearson correlation coefficient and spectral saturation levels.

(5) Finally, features are sorted by the PS values in descending order; the smaller the PS value, the higher the importance of the feature.

### 3.4. Model Evaluation and Application

To evaluate the capability of the proposed feature evaluation criteria, the saturation levels of all features were derived from the kriging model with a spherical function and the quadratic model, and optimal feature sets were finally obtained with the Pearson correlation coefficient and the proposed feature selection criteria. Next, four models (SVM, MLR, RF

and KNN [41,42,47]) were applied to map the forest GSV using the optimal feature set. The flowchart of mapping GSV is illustrated in Figure 4.

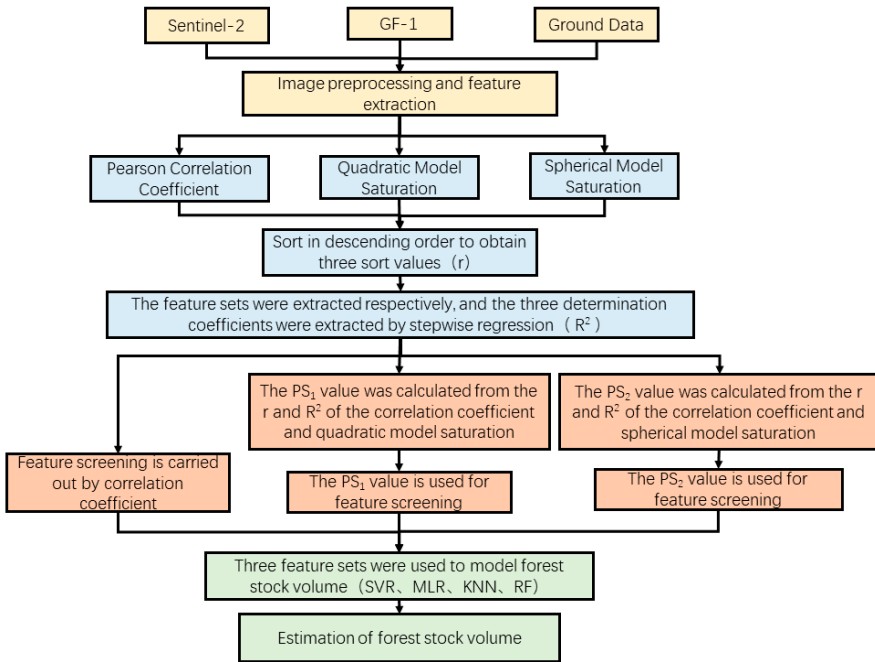

**Figure 4.** The framework of forest GSV estimation with new feature evaluation criteria based on spectral saturation.

Additionally, the accuracy of the mapped GSV was evaluated with leave-one-out cross-validation (LOOCV). The adjusted determination coefficient ($R^2$), root mean square error (*RMSE*), and relative root mean square error (*rRMSE*) between the predicted and ground-measured GSV were selected as the evaluation indices [48].

$$R^2 = 1 - \frac{\left(1 - \left[1 - \frac{\sum_{i=1}^{n}(y_i - \hat{y}_i)^2}{\sum_{i=1}^{n}(y_i - \overline{y_i})^2}\right]^2\right)(n-1)}{n - p - 1} \tag{8}$$

$$RMSE = \sqrt{\frac{\sum_{i=1}^{n}(y_i - \hat{y}_i)^2}{n}} \tag{9}$$

$$rRMSE = \frac{RMSE}{\overline{y_i}} \times 100\% \tag{10}$$

where $y_i$ is the measured GSV, $\hat{y}_i$ is the predicted GSV, $\overline{y_i}$ is the average of the measured GSV, and $n$ is the number of samples.

## 4. Results

### 4.1. Saturation Values of Features

To evaluate the sensitivity between GSV and features, the saturation levels of alternative spectral features should first be estimated. In our study, the quadratic model and the spherical model were employed to estimate the saturation values of the features extracted from GF-1 and Sentinel-2 images, respectively (Figure 5 and Table 3). Figure 5 illustrates the fitted curves of the top ten features extracted from GF-1 (Figure 5a,b) and Sentinel-2 (Figure 5c,d) images. For one feature, the curves fitted by the quadratic model and the spherical model were basically the same, and the estimated values of spectral saturation were slightly different. However, the fitted curves and their saturation levels were significantly different for different remote sensing images.

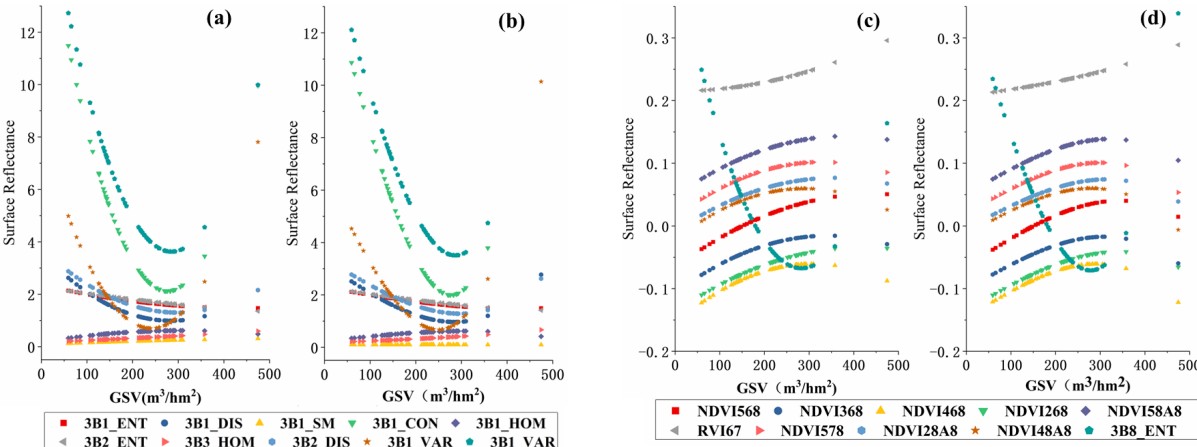

**Figure 5.** The fitted curves of the top ten features extracted from GF-1 images (**a**,**b**) and Sentinel-2 images (**c**,**d**) using the quadratic model and the spherical model, respectively.

**Table 3.** Spectral saturation levels of the top ten feature variables using two models for different images.

| GF-1 | Correlation Coefficient | Quadratic Model (m³/hm²) | Spherical Model (m³/hm²) | Sentinel-2 | Correlation Coefficient | Quadratic Model (m³/hm²) | Spherical Model (m³/hm²) |
|---|---|---|---|---|---|---|---|
| 3B1_ENT | −0.51 | 446.60 | 417.60 | NDVI568 | 0.50 | 444.02 | 343.82 |
| 3B1_DIS | −0.48 | 284.28 | 285.21 | NDVI368 | 0.49 | 342.89 | 309.16 |
| 3B1_SM | 0.48 | 474.54 | 474.54 | NDVI468 | 0.48 | 309.41 | 291.99 |
| 3B1_CON | −0.47 | 276.12 | 277.37 | NDVI268 | 0.47 | 418.92 | 340.28 |
| 3B1_HOM | 0.45 | 307.18 | 307.15 | NDVI58A8 | 0.47 | 382.43 | 322.14 |
| 3B2_ENT | −0.43 | 474.54 | 474.54 | RVI67 | 0.46 | 474.54 | 474.54 |
| 3B3_HOM | 0.42 | 474.54 | 474.54 | NDVI578 | 0.45 | 330.82 | 301.85 |
| 3B2_DIS | −0.41 | 298.22 | 297.90 | NDVI28A8 | 0.45 | 357.96 | 314.99 |
| 3B1_VAR | −0.41 | 240.78 | 249.22 | NDVI48A8 | 0.43 | 289.14 | 279.69 |
| 3B2_CON | −0.40 | 285.45 | 286.76 | 3B8_ENT | −0.42 | 283.17 | 278.36 |

For the top ten features, the absolute values of the Pearson correlation coefficient ranged from 0.45 and 0.65 for the features extracted from GF-1 images and from 0.40 to 0.51 for the features extracted from Sentinel-2 images. Moreover, for the GF-1 images, the saturation levels ranged from 240.78 m³/hm² to 474.54 m³/hm² using the quadratic model and from 249.22 m³/hm² to 474.54 m³/hm² using the spherical model. For the Sentinel-2 images, the absolute values of the Pearson correlation coefficient between the extracted features and forest GSV ranged from 0.42 to 0.50, and the spectral saturation levels ranged from 283.17 m³/hm² to 474.54 m³/hm² using the quadratic model and from 278.36 m³/hm² to 474.54 m³/hm² using the spherical model. It was also found that the orders of spectral saturation did not match the orders of the Pearson correlation coefficient. When using the Pearson correlation to select features, the results proved that the optimal feature set ignored the saturation effect. Therefore, it was necessary to retrieve the optimal feature set based on the Pearson correlation and saturation levels.

### 4.2. The Results of Proposed Feature Evaluation Criteria

To obtain the initial feature set, all extracted features were first sorted in descending order according to Pearson correlation and spectral saturation levels. Then, the linear stepwise regression model was employed to calculate the adjusted determination coefficient ($R^2$) for various sorted features. Ultimately, the initial feature set was constructed from the union of the two feature sets extracted from the Pearson correlation and spectral saturation levels w the maximum determination coefficient. In this study, based on Pearson correlations, the saturation level of the quadratic model, and the saturation level of the

spherical model, the maximum determination coefficients were 0.412, 0.322, and 0.303 for the GF-1 images and 0.401, 0.302, and 0.319 for the Sentinel-2 images, respectively. Using the proposed feature evaluation criteria, the new importance evaluation index (PS) of the selected features in the initial feature set was constructed using the serial numbers and determination coefficients. The top ten PS values extracted from the GF-1 and Sentinel-2 image sets are listed in Tables 4 and 5, respectively.

**Table 4.** PS values of top 10 variables extracted from GF-1 images.

| NO. | Pearson ($r_1$) and Quadratic Model ($r_2$) | | | | Pearson ($r_1$) and Spherical Model ($r_2$) | | | |
|---|---|---|---|---|---|---|---|---|
| | Variable | $PS_1$ | $r_1$ | $r_2$ | Variable | $PS_2$ | $r_1$ | $r_2$ |
| 1 | 3B1_ENT | 17.96 | 1 | 5 | 3B1_ENT | 18.93 | 1 | 5 |
| 2 | 3B1_SM | 19.70 | 3 | 4 | 3B3_HOM | 20.29 | 7 | 1 |
| 3 | 3B3_HOM | 20.10 | 7 | 1 | 3B1_SM | 20.48 | 3 | 4 |
| 4 | 3B2_ENT | 20.77 | 6 | 2 | 3B2_ENT | 24.46 | 6 | 3 |
| 5 | 3B1_HOM | 33.88 | 5 | 7 | 3B1_HOM | 35.24 | 5 | 7 |
| 6 | 3B1_DIS | 35.91 | 2 | 10 | 3B1_DIS | 37.86 | 2 | 10 |
| 7 | 3B2_SM | 43.30 | 14 | 3 | 3B2_SM | 40.58 | 14 | 2 |
| 8 | 3B1_CON | 43.87 | 4 | 11 | 3B2_DIS | 45.82 | 8 | 8 |
| 9 | 3B2_DIS | 44.26 | 8 | 8 | 3B1_CON | 46.01 | 4 | 11 |
| 10 | 3B2_HOM | 45.33 | 11 | 6 | 3B2_HOM | 46.50 | 11 | 6 |

**Table 5.** PS values of the top 10 variables extracted from Sentinel-2 images.

| NO. | Pearson ($r_1$) and Quadratic Model ($r_2$) | | | | Pearson ($r_1$) and Spherical Model ($r_2$) | | | |
|---|---|---|---|---|---|---|---|---|
| | Variable | $PS_1$ | $r_1$ | $r_2$ | Variable | $PS_2$ | $r_1$ | $r_2$ |
| 1 | NDVI568 | 15.74 | 1 | 4 | NDVI568 | 15.03 | 1 | 4 |
| 2 | RVI67 | 24.90 | 6 | 3 | RVI67 | 24.37 | 6 | 3 |
| 3 | NDVI268 | 26.53 | 4 | 5 | NDVI268 | 25.65 | 4 | 5 |
| 4 | NDVI368 | 31.48 | 2 | 8 | NDVI368 | 30.07 | 2 | 8 |
| 5 | NDVI58A8 | 32.34 | 5 | 6 | NDVI58A8 | 31.28 | 5 | 6 |
| 6 | NDVI5128 | 33.24 | 12 | 1 | NDVI5128 | 33.06 | 12 | 1 |
| 7 | NDVI5118 | 34.05 | 11 | 2 | NDVI5118 | 33.70 | 11 | 2 |
| 8 | NDVI468 | 40.59 | 3 | 10 | NDVI468 | 38.83 | 3 | 10 |
| 9 | NDVI28A8 | 43.13 | 8 | 7 | NDVI28A8 | 41.89 | 8 | 7 |
| 10 | NDVI578 | 47.26 | 7 | 9 | NDVI578 | 45.67 | 7 | 9 |

When using the proposed feature evaluation criteria, the PS values of the top 10 variables extracted from the GF-1 images ranged from 17.96 to 45.33 with the quadratic model and from 18.93 to 46.50 with the spherical model. Compared with a single criterion, the PS values between the feature and GSV also contained contributions from the Pearson correlation coefficient and spectral saturation levels. Therefore, the PS values were determined by the orders of the Pearson correlation coefficient and saturation levels. It is illustrated that some features with high Pearson correlation coefficients were excluded from the top 10 variables extracted with the proposed method because of low saturation levels, such as 3B1_VAR extracted from the GF-1 images. Moreover, the values of PS were also related to the employed models used to extract spectral saturation levels. The gaps of estimated spectral saturation levels were mainly induced by the quadratic and spherical models. When using the new criteria, most of the variables were in the top ten, but the orders changed.

For the Sentinel-2 images (Table 5), the orders of the top ten variables evaluated with the proposed method were rather different from those of the variables extracted with Pearson correlation coefficients or spectral saturation levels. After using the proposed method, the ranks of the features changed. Though there were some differences in the estimated saturation values when using the different estimation models, it was found that

these models had no effect on the rank of the top ten features when using the proposed feature selection method.

### 4.3. Optimal Feature Set

Using the linear stepwise regression model and determination coefficients, the initial feature set was formed from the union of the two feature sets extracted from the Pearson correlations and spectral saturation levels with the maximum determination coefficient. Then, the PS values of all initial feature sets were retrieved with the proposed feature selection method. To guarantee the independence of the features, variance inflation factor (VIF) was introduced to conduct collinearity analysis, and refined feature sets were constructed after removing linearly dependent features. Finally, after using linear stepwise regression models, optimal feature sets of various feature selection methods were obtained from the refined feature set in descending order. In our study, three optimal feature sets (Table 6) were derived using Pearson correlation coefficients, PS values with the spherical model, and PS values with the quadratic model. For various feature selection methods, the numbers of features contained in the optimal feature set are largely dependent on the feature selection methods.

**Table 6.** Optimal feature set according to various feature selection methods.

| Data | Feature Selection Method | Optimal Feature Set |
|---|---|---|
| GF-1 | Pearson (5) | 3B1_ENT, 3B1_SM, 3B4_ENT, 3B4_HOM, 3B4_SM |
| | Pearson and spherical model (7) | 3B1_ENT, 3B1_DIS, 3B1_SM, 3B1_CON, 3B1_HOM, 3B2_ENT, 3B3_HOM |
| | Pearson and quadratic model (7) | 3B1_ENT, 3B1_DIS, 3B1_SM, 3B1_CON, 3B1_HOM, 3B2_ENT, 3B3_HOM |
| Sentinel-2 | Pearson (7) | NDVI568, NDVI368, NDVI6128, NDVI285, 3B8_ENT, 3B8_SM, RVI85 |
| | Pearson and spherical model (4) | 3B8_ENT, 3B8_SM, NDVI5118, NDVI5128 |
| | Pearson and quadratic model (4) | 3B8_ENT, 3B8_SM, NDVI5118, NDVI5128 |

For the GF-1 images, the optimal feature set mainly came from texture information, and the number of feature sets was dependent on the FS methods. There were five features in the optimal feature set using Pearson correlations and seven features in the optimal feature set using the proposed method. Furthermore, these optimal feature sets with different models used for estimating saturation levels had the same features. For the Sentinel-2 images, most selected features came from vegetation indices, and the results of the optimal feature set were the same when using different models to estimate the saturation levels. It was inferred that the optimal feature sets obtained from the proposed method were largely dependent on the relative changes in saturation levels.

### 4.4. The Results of Estimated Forest GSV

To further analyze the capability of the proposed feature evaluation criteria, four models (SVM, MLR, RF and KNN) were employed to estimate forest GSV using two optimal feature sets extracted from GF-1 and Sentinel-2 images. LOOCV was used to retrieve the evaluation indices, including the determination coefficient ($R^2$), root mean square error (RMSE), and relative root mean square error (rRMSE) between the predicted and measured GSV values. The estimated results of the four models are listed in Table 7.

Table 7 shows that the proposed feature evaluation criteria could effectively improve the accuracy of estimating forest GSV. When using the optimal feature set obtained with Pearson correlations, the rRMSE values ranged from 32.57% to 34.82% for the GF-1 images

and from 30.64% to 36.53% for the Sentinel-2 images. After using the optimal feature set obtained with the proposed feature evaluation criteria, the saturation levels and Pearson correlations between features and GSV were employed to evaluate the capability of the features in mapping GSV. The rRMSE values ranged from 28.67% to 33.08% for the GF-1 images and from 26.85% to 33.28% for the Sentinel-2 images. The highest GSV mapping accuracy was obtained with the RF model for the GF-1 images ($R^2$ = 0.49, rRMSE = 28.67%) and with the SVM model for the Sentinel-2 images ($R^2$ = 0.52, rRMSE = 26.65%). It was obvious that the optimal feature set obtained with the proposed method was better able to map GSV than that obtained with Pearson correlation coefficients.

**Table 7.** GSV estimated with various feature selection methods.

| Criteria | Model | GF-1 | | | Sentinel-2 | | |
|---|---|---|---|---|---|---|---|
| | | $R^2$ | RMSE (m³/hm²) | rRMSE (%) | $R^2$ | RMSE (m³/hm²) | rRMSE (%) |
| Pearson correlation | MLR | 0.43 | 65.13 | 32.96 | 0.24 | 73.96 | 35.87 |
| | KNN | 0.32 | 70.51 | 34.19 | 0.31 | 71.03 | 34.95 |
| | SVM | 0.45 | 64.23 | 32.57 | 0.38 | 63.17 | 30.64 |
| | RF | 0.29 | 71.79 | 34.82 | 0.23 | 82.72 | 36.53 |
| Proposed method | MLR | 0.44 | 62.11 | 30.91 | 0.38 | 74.91 | 33.28 |
| | KNN | 0.38 | 66.48 | 33.08 | 0.47 | 58.09 | 28.17 |
| | SVM | 0.47 | 60.82 | 30.27 | 0.52 | 55.62 | 26.65 |
| | RF | 0.49 | 58.67 | 28.67 | 0.45 | 62.22 | 30.18 |

To further analyze the estimated forest GSV, scatter plots between ground-measured and predicted GSV were plotted using the GF-1 images (Figure 6) and the Sentinel-2 images (Figure 7). It was found that the accuracy of mapping forest GSV relied on the feature evaluation criteria. After using the proposed method, the features were simultaneously evaluated by double criteria, saturation levels and Pearson correlation, and overestimated results with low GSV values were significantly improved by adding some features with high saturation levels to the optimal feature set. Figures 6 and 7 illustrate that the number of underestimated samples was significantly decreased after using the proposed feature selection method (Figures 6e–h and 7e–h). Therefore, considering the spectral saturation levels of the features, the accuracy of estimated GSV was sufficiently improved by expanding the range of GSV. However, due to the limited capability of optical images, saturation still occurred for the samples with a GSV of more than 300 m³/hm².

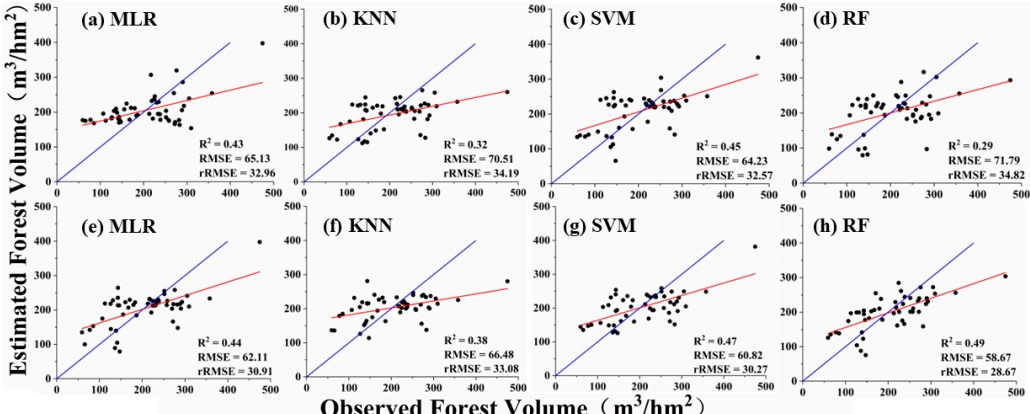

**Figure 6.** Scatter plots between measured and predicted GSV with different feature selection methods and models using GF-1 images; (**a–d**) show the results using Pearson correlation and (**e–h**) show the results using the proposed method.

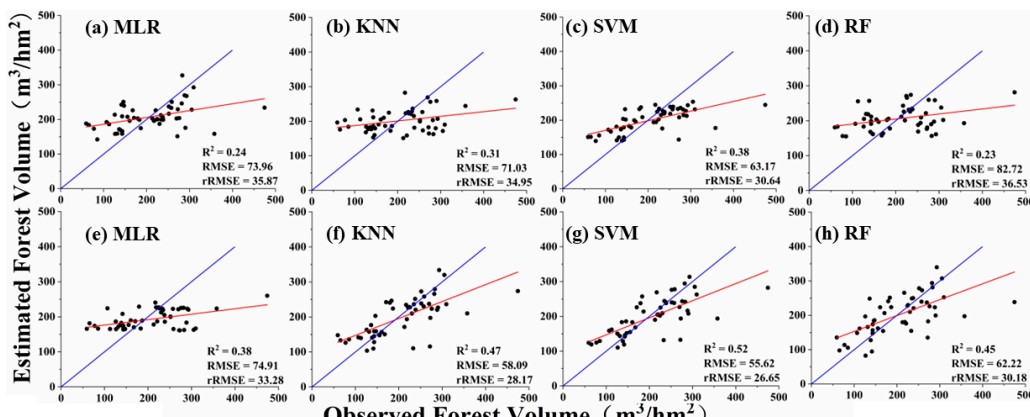

**Figure 7.** Scatter plots between measured and predicted GSV with different feature selection methods and models using Sentinel-2 images; (**a**–**d**) show the results using Pearson correlation and (**e**–**h**) show the results using the proposed method.

To describe the gaps between different feature criterion methods, two models (SVM and RF) were employed to map the planted Chinese fir forest using the GF-1 and Sentinel-2 images (Figure 8). When using the optimal feature set obtained with the proposed method, the estimated GSV was obviously larger than that obtained with the Pearson criterion (Figure 8e–h). The gaps of the estimated GSV with different feature selection methods were caused by the capabilities of the selected features. These results (Figure 9) illustrate that the ranges of the estimated GSV were extended after adding some features with high saturation levels. For a forest with a low GSV (less than 100 m$^3$/hm$^2$), overestimated results often occur when using an optimal feature set obtained with the Pearson criterion. After using the proposed method, the proportion of low GSV values (less than m$^3$/hm$^2$) decreased from 13.00% to 3.37% for the GF-1 images and from 15.53% to 4.08% for the Sentinel-2 images. Meanwhile, the proportion of high GSV (ranging from 200 m$^3$/hm$^2$ to 300 m$^3$/hm$^2$) increased from 42.49% to 47.16% for the GF-1 images and from 40.47% to 50.86% for the Sentinel-2 images. It can be inferred that the sensitivity of the optimal feature set and the accuracy of estimated GSV could be improved by viewing saturation levels as an evaluation criterion. However, once a forest's GSV is larger than its saturation levels, the saturation phenomenon inevitably occurs in optical images.

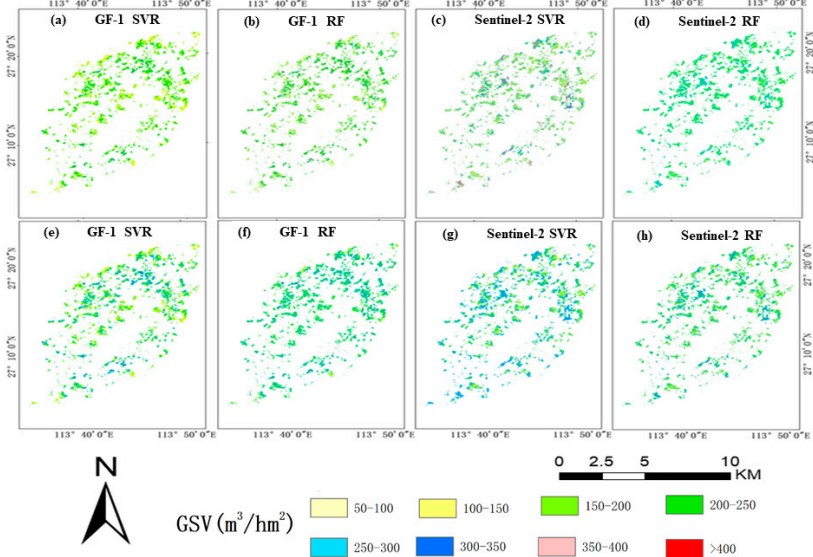

**Figure 8.** Map of forest GSV with SVM and RF from GF-1 and Sentinel-2 images. (**a**–**d**) show the results using Pearson correlation and (**e**–**h**) show the results using the proposed method.

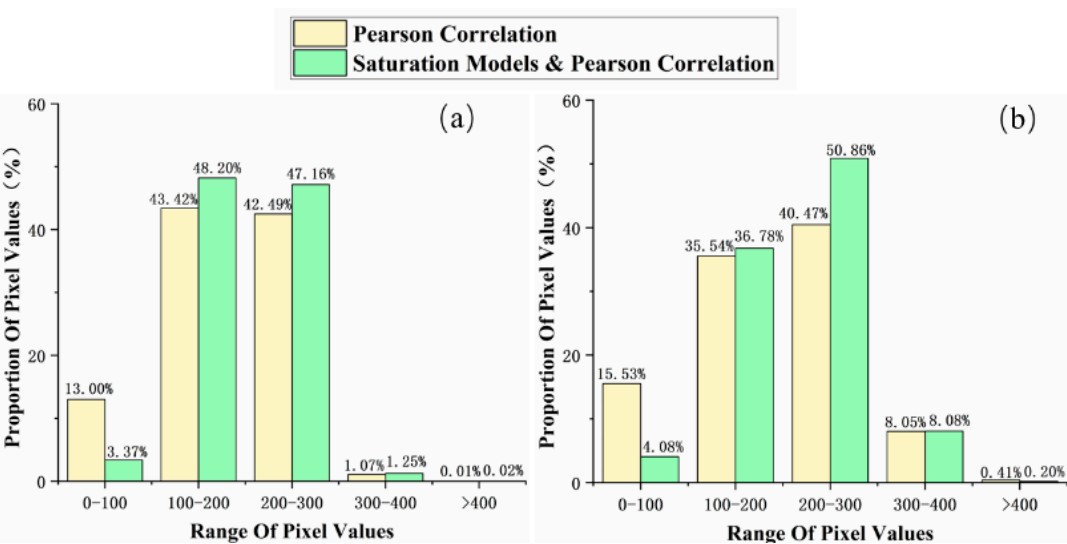

**Figure 9.** Histograms of estimated GSV, (**a**) derived from RF for GF-1 images and (**b**) from SVM for Sentinel-2 images.

## 5. Discussion

### 5.1. Saturation Levels and Quantitative Model

Optical images are widely used for mapping forest parameters, such as AGB and GSV [12,18,23,32]. Additionally, spectral variables are insensitive to changes in GSV when the GSV or AGB is larger than certain levels. Consequently, it is rather important to construct a quantitative model for evaluating the capability of mapping GSV for alternative features. Essentially, quantitative models of saturation levels can be employed to express the relationships between remote sensing features and AGB, and the extreme values or trends of the models regarded as saturation levels can be determined with a mathematic method. At present, several parameter models have been used to determine the saturation levels of alternative features extracted from various images and forest types, such as the semi-exponential, kriging, and quadratic models [18,22]. In our study, the kriging model with a spherical function and the quadratic model were employed to obtain fitted curves, and the extreme points of the fitted curves were regarded as the saturation levels for each feature.

Furthermore, the quantitative model of saturation levels is different from the estimation models of forest GSV. Normally, the objective of a quantitative model of saturation levels is to seek a unique extreme point of fitted curves, and a minimum of error between estimated and measured GSV is regarded as the goal for estimation models of forest GSV. The extreme points of fitted curves depend on the selected models and observations. Here, it was confirmed that the estimated saturation level was largely dependent on the quantitative model for a certain feature. Figure 6 and Table 4 illustrate the saturation levels obtained with the two models. For most features, the difference in the saturation levels obtained with the two quantitative models was less than 50 $m^3/hm^2$, and there were few variables with roughly the same estimated saturation levels.

Additionally, the disadvantages of the kriging model with a spherical function and the quadratic model are that the shapes of their fitted curves are unreasonable when the GSV is larger than extreme points (saturation levels), and the gaps between the fitted curves and ground-measured GSV grow larger as the forest GSV increases. It was found that these gaps are caused by the characteristics of the kriging model with a spherical function and the quadratic model. Concretely, these fitted curves have the capability for the accurate description the relationships between GSV and features when the GSV is lower than the saturation level. Once the GSV is larger than the saturation level, it is very hard to express the relationships between features and the forest GSV for these models.

### 5.2. The Contribution of Proposed Feature Selection Method

For a planted forest with a high GSV, the saturation phenomenon is considered an important factor that limits the accuracy of mapping the GSV using optical images. Previous studies have shown that spectral saturation problems are caused by various factors, such as remote sensing images, forest type, stand parameters, and topography, especially for forests with dense canopies [32]. Recently, researchers have proposed several potential solutions to solve the spectral saturation problem, such as using the forest height (CHM) derived from various methods and using time-series optical images [22,32]. Additionally, previous studies also have shown that saturation levels can be delayed by selecting appropriate remote sensing variables [2,23,33]. Normally, optimal feature sets are dependent on the feature selection methods. The accuracy of mapping forest AGB or GSV has been increased to some extent through various improved feature selection methods based on filters, wrappers, and embedded models. Simultaneously, delayed saturation levels can enlarge the range of estimated GSV with the use of selection variable sets with high saturation values. However, these saturation values are rarely directly viewed as an evaluation criterion used to select features.

Essentially, the key to improve the accuracy of estimated GSV is that the selected features have high saturation levels. In our study, the proposed feature selection method based on the saturation values of alternative variables was constructed to obtain optimal feature sets. The saturation levels and Pearson correlation coefficients between the features and GSV were employed together to evaluate the ability of features for mapping GSV. The results showed that the number of overestimated samples and errors between the ground-measured and estimated GSV were obviously decreased after using the proposed feature selection methods, especially for the samples with a GSV of less than 300 $m^3/hm^2$ (Figures 8 and 9).

To further demonstrate the bias of the mapped results, the errors between the ground-measured and predicted GSV using two feature criteria are illustrated in Figures 10 and 11. Generally, the errors between the predicted and ground-measured GSV presented a specific distribution for mapping forest GSV using optical remote sensing images. Figures 10 and 11 illustrate that the results of our study using Pearson correlation were basically consistent with those of previous studies for four models and that the rRMSE values ranged from 30% to 40% [2,23,33]; this consistency is also reflected in the underestimated results in the high GSV images and the overestimation results in the low GSV images [23,33]. Therefore, the accuracy of mapping forest GSV is largely constrained by these overestimation and underestimation phenomena when using traditional feature criteria. The major reason for this is that the signals of optical remote sensing images without penetration capture changes in forest GSV in terms of the forest canopy [18,23]. In young forests (less than 100 $m^3/hm^2$), the canopy increased faster than the forest GSV and over-estimated frequently occurred. In mature or over-mature forests, the forest canopy increased slower than the forest GSV and the saturation phenomenon often occurred.

In our study, the capability of features was double-evaluated with the proposed feature selection method based on the spectral saturation and Pearson correlation coefficients. Then, the systematic distribution of residuals was broken by using the optimal feature set obtained from the proposed method (Figures 10e–h and 11e–h). The results showed that the number of overestimated samples was significantly decreased after adding some features with high saturation levels. It was found that the improved forest GSV ranged from 100 $m^3/hm^2$ to 300 $m^3/hm^2$. For the samples with a GSV smaller than 100 $m^3/hm^2$ or larger than 300 $m^3/hm^2$, the contribution of the proposed method was very weak (Figures 10 and 11). The saturation phenomenon remains an inevitable issue for mapping forest GSV or ABG using optical remote sensing images. To overcome the saturation phenomenon, vertical features of forests, such as the canopy height model (CHM) or features extracted from polarimetric SAR images, are regarded as an effective means to delay the saturation problem [15,16,22]. In addition, the bias of mapped GSV is also affected by the errors of ground-measured GSV involved in several forest parameters

(DBH and H), as well as the uncertainty of the binary volume table of planted Chinese fir. However, compared with the recently estimated accuracy and bias in mapping forest GSV using remote sensing images, the influence of ground-measured errors on the accuracy of estimation is much smaller than the errors induced by remote sensing images and models. Therefore, the uncertainty of mapped forest GSV should be further evaluated after obtaining reliable features and models.

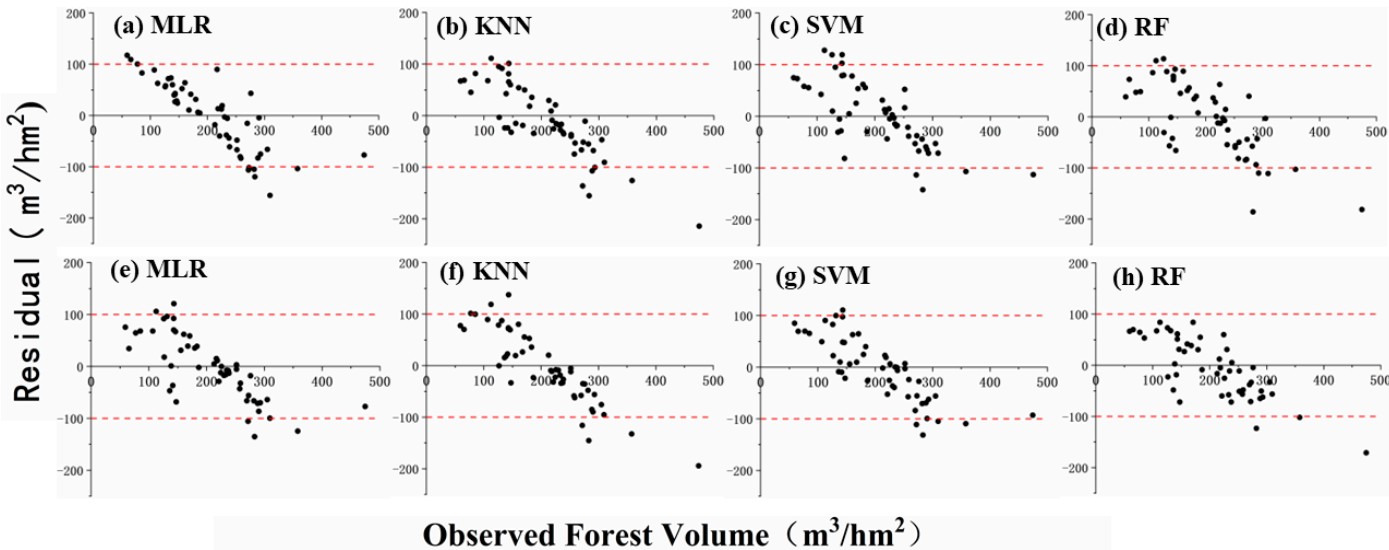

**Figure 10.** Scatter plots of errors between ground-measured and predicted GSV with different feature selection methods and models; (**a**–**d**) show results from GF-1 images using Pearson correlation and (**e**–**h**) show the results from GF-1 images using the proposed method.

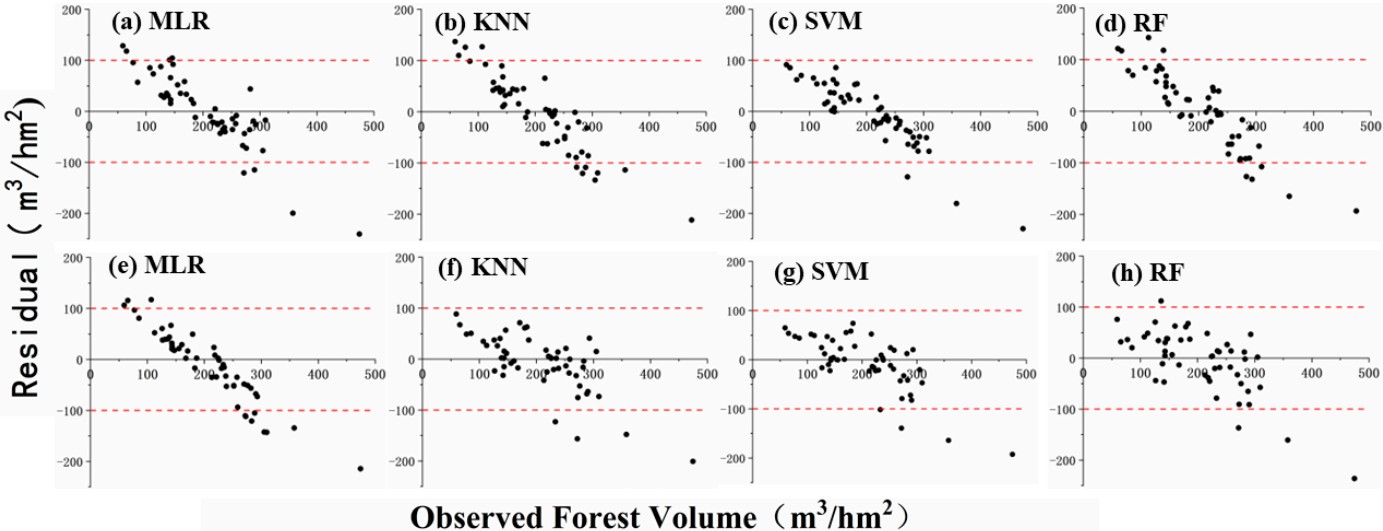

**Figure 11.** Scatter plots of errors between ground-measured and predicted GSV with different feature selection methods and models; (**a**–**d**) show results from Sentinel-2 images using Pearson correlation and (**e**–**h**) show the results from Sentinel-2 images using the proposed method.

## 6. Conclusions

In this study, the saturation levels of remote sensing features derived from GF-1 and Sentinel-2 images were estimated with different quantitative models. Then, novel criteria for evaluating features were constructed based on the saturation levels and Pearson correlation coefficients between features and GSV. The optimal feature sets were obtained with the proposed criteria for evaluating features and the linear stepwise regression model. After

mapping forest GSV with four widely used machine learning models, the results illustrated that the proposed feature evaluation method as better able to select an optimal feature set for mapping GSV. It was concluded that the sensitivity of the optimal feature set and the accuracy of estimated GSV could be improved by viewing saturation levels as an evaluation criterion. However, the contribution of the proposed method was found to be very weak for forests with a GSV of more than 300 m$^3$/hm$^2$. The saturation phenomenon remains an inevitable issue when mapping forest GSV or ABG using optical remote sensing images. In the future, the accuracy of mapping forest GSV will be improved using spaceborne LiDAR (GEDI and ICESat-2) products to delay the saturation levels of features, and multi-temporal optical remote sensing images will be also applied to improve the sensitivity of features.

**Author Contributions:** Conceptualization, H.L., J.L. and Z.L.; methodology, J.L. and Z.L.; software, W.Z., Z.L. and Z.Y.; validation, H.L., J.L. and Z.L.; formal analysis, W.Z., P.Y., T.Z. and Z.L.; investigation, H.L., J.L., Z.L., Z.Y. and P.Y.; resources, H.L., W.Z., J.L. and P.Y.; data processing, W.Z., Q.W., H.R.M. and P.Y.; original draft, W.Z.; review and revision, H.L., J.L. and W.Z.; final editing: J.L.; visualization, J.L. and W.Z.; supervision, H.L. and J.L.; project administration, H.L. and J.L.; funding acquisition, H.L., J.L. and Z.L. All authors have read and agreed to the published version of the manuscript.

**Funding:** This research was funded by the National Natural Science Foundation of China (Project number: 32171784), the Hunan Provincial Natural Science Foundation of China (Project number: 2021JJ31158), the Excellent Youth Project of the Scientific Research Foundation of the Hunan Provincial Department of Education (Project number: 21B0246), and the postgraduate scientific research Innovative project of Hunan province (Project number: CX20210854).

**Data Availability Statement:** The observed GSV data from the sample plots and the spatial distribution data of forest resources presented in this study are available on request from the corresponding author. Those data are not publicly available due to privacy and confidentiality. The Sentinel-2 and GF-1 stereo data were obtained from the US Geological Survey Earth Explorer website (http://earthexplorer.usgs.gov/, accessed on 11 June 2021).

**Conflicts of Interest:** The authors declare no conflict of interest.

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
