# Peer review of "Mapping Forest Growing Stem Volume Using Novel Feature Evaluation Criteria Based on Spectral Saturation in Planted Chinese Fir Forest"

_remotesensing, doi:10.3390/rs15020402_

Round 1

Reviewer 1 Report

Thank you for being able to review this manuscript. This study mapping forest growing stem volume using a novel feature evaluation criterion based on spectral saturation. The topic is interesting, but several issues should be considered before publication:

Abstract

Line 20 - “depends on the saturation levels of features derived from optical remote sensing images”. Does it just depend on that?

Introduction

Line 47 - “During the past ten years”. And before that period?

Line 71 - “LiDAR is not suitable”. Land dealing (TLS, MLS, or PLS) or air? What about orbital LiDAR data, such as from the GEDI and ICESat-2 missions? What about 3D data derived from DAP-UAV? Please rewrite this sentence.

What is the scientific contribution of this work? What is the research gap addressed by this study?

Study Area and Data

data preparation

Line 129 - What are the collection months in 2016 and 2017? Were they the same as the GF-1 and Sentinel-2 data collection? If not, this could be a problem. Explain.

Line 130 - How many parcels of 20 x 20? and 30 x 30? inform.

When fitting the model (volume as a function of optical data), could this difference in size cause problems? Explain this in the text.

Line 132 - “accurately measured”. Enter the general RMSE for X, Y, and Z of these points.

Line 133 - “several parameters”. There weren't many, just 4.

How was the height estimated? A ruler? Clinometer, describe.

How was the DBH estimated? To describe.

Crown diameter? Canopy density? To describe.

What were the height and DBH measurement error? This is important for this type of study. Was this measurement error propagated in the final models using optical data?

Line 136 - “Cunninghamia lanceolata plantation in Hunan province”. What would that be?

It is important to present the volume equation and how it was adjusted. What is the error in the adjustment? Was this error propagated when estimating the volume with optical data? This needs to be inserted into the text. Remembering that the purpose of the article is to estimate the estimate, that is, to estimate the estimated volume (by a volume equation) via optical data.

How was the volume integrated into the hectare?

Figure 2b graph does not show volume 0 m³/ha. Explain this difference to the text (line 139).

 Remote sensing data

What are the wavelengths of each band (GF-1 and Sentinel 2)? Inform. How many scenes were used for each satellite, just one? Inform.

Pre-processing: describe each of these steps, which methods, and how they were carried out.

Line 154 - “reduce the matching error”... how and by how much was this error reduced? Has any analysis been performed to assess this error? Inform.

What is the problem with resampling a 60 m image to 30 m? Explain.

Methods

Table 1 - Enter IVs references.

Line 211 - m³/hm²? What does that mean? Review all manuscript.

Results

Figure 1 - Confused. Improve.

What does 3B1_ENT, NDVI 568 mean? The reader is not obliged to go back into the text or guess these acronyms.

Optimal Feature Set

Line 330 to 342 - Must be inserted in the methodology.

The results of estimated forest GSV

Table 5

Would it be fair to compare the Pearson model (GF-1), with 5 variables, with the Pearson & Spherical model, with 7 variables? Better performance is expected from models with more explanatory variables. I suggest comparing models with the same number of explanatory variables. I suggest inserting it in the manuscript.

What are the VIF values for Pearson's Sentinel model - 2 with 4 different NDVIs, < 10? Present/discuss this result.

Figure 8, Sentinel - 1?

By analyzing Figures 7 and 8, it is not possible to notice improvements. Graphs 7f and 8e, for example, clearly show the saturation of the optical data to estimate the highest volume values. Explain these results.

Reviewer 2 Report

Please find the pdf in the attachment

Round 2

Reviewer 1 Report

Thank you for reviewing this manuscript. Here are some important considerations.

Since height and DBH errors are negligible, report the values in the body of the manuscript. Also insert in the discussion of the text that these errors were not considered when adjusting the models with remote sensing data and what the possible implications are.

Insert the wood volume estimation equation and its error.

Also, inform in the discussion, the possible problems of not propagating this error when fitting the model with remote sensing data.

Thanks.

Reviewer 2 Report

Accept in the current form

Author Response

Thank you for the review.